# Unpaired Point Cloud Completion on Real Scans using Adversarial Training

**Xuelin Chen**
Shandong University
University College London

**Baoquan Chen**
Peking University

**Niloy J. Mitra**
University College London
Adobe Research London

## Abstract

As 3D scanning solutions become increasingly popular, several deep learning setups have been developed for the task of scan completion, i.e., plausibly filling in regions that were missed in the raw scans. These methods, however, largely rely on supervision in the form of paired training data, i.e., partial scans with corresponding desired completed scans. While these methods have been successfully demonstrated on synthetic data, the approaches cannot be directly used on real scans in the absence of suitable paired training data. We develop a first approach that works directly on input point clouds, does not require paired training data, and hence can directly be applied to real scans for scan completion. We evaluate the approach qualitatively on several real-world datasets (ScanNet, Matterport3D, KITTI), quantitatively on 3D-EPN shape completion dataset, and demonstrate realistic completions under varying levels of incompleteness.

## 1 Introduction

Robust, efficient, and scalable solutions now exist for easily scanning large environments and workspaces (Dai et al., 2017a; Chang et al., 2017). The resultant scans, however, are often partial and have to be completed (i.e., missing parts have to be hallucinated and filled in) before they can be used in downstream applications, e.g., virtual walk-through, path planning.

The most popular data-driven scan completion methods rely on *paired supervision* data, i.e., for each incomplete training scan, a corresponding complete data (e.g., voxels, point sets, signed distance fields) is required. One way to establish such a shape completion network is then to train a suitably designed encoder-decoder architecture (Dai et al., 2017b; 2018). The required paired training data is obtained by virtually scanning 3D objects (e.g., SunCG Song et al. (2017), ShapeNet Chang et al. (2015) datasets) to simulate occlusion effects. Such approaches, however, are unsuited for real scans where large volumes of paired supervision data remain difficult to collect. Additionally, when data distributions from virtual scans do not match those from real scans, completion networks trained on synthetic-partial and synthetic-complete data do not sufficiently generalize to real (partial) scans. To the best of our knowledge, no point-based unpaired method exists that learns to translate noisy and incomplete point cloud from raw scans to clean and complete point sets.

We propose an unpaired point-based scan completion method that can be trained *without* requiring explicit correspondence between partial point sets (e.g., raw scans) and example complete shape models (e.g., synthetic models). Note that the network does *not* require explicit examples of real complete scans and hence existing (unpaired) large-scale real 3D scan (e.g., Dai et al. (2017a); Chang et al. (2017)) and virtual 3D object repositories (e.g., Song et al. (2017); Chang et al. (2015)) can directly be leveraged as training data. Figure 1 shows example scan completions. As we show in Table 1, unlike methods requiring paired supervision, our method continues to perform well even if the data distributions of synthetic complete scans and real partial scans differ.

We achieve this by designing a generative adversarial network (GAN) wherein a generator, i.e., an adaptation network, transforms the input into a suitable latent representation such that a discriminator cannot differentiate between the transformed latent variables and the latent variables obtained from training data (i.e., complete shape models). Intuitively, the generator is responsible for the key task of mapping raw partial point sets into clean and complete point sets, and the process is regu-

Figure 1: We present a point-based shape completion network that can be directly used on raw scans without requiring paired training data. Here we show a sampling of results from the ScanNet, Matterport3D, 3D-EPN, and KITTI datasets.

larized by working in two different latent spaces that have separately learned manifolds of scanned and synthetic object data.

We demonstrate our method on several publicly available real-world scan datasets namely (i) Scan-Net (Dai et al., 2017a) chairs and tables; (ii) Matterport3D (Chang et al., 2017) chairs and tables; and (iii) KITTI (Geiger et al., 2012) cars. In absence of completion ground truth, we cannot directly compute accuracy for the completed scans, and instead compare using plausibility scores. Further, in order to quantitatively evaluate the performance of the network, we report numbers on a synthetic dataset (Dai et al., 2017b) where completed versions are available. Finally, we compare our method against baseline methods to demonstrate the advantages of the proposed unpaired scan completion framework.

## 2 Related Work

**Shape Completion.**     Many deep neural networks have been proposed to address the shape completion challenge. Inspired by CNN-based 2D image completion networks, 3D convolutional neural networks applied on voxelized inputs have been widely adopted for 3D shape completion task (Dai et al., 2018; 2017b; Sharma et al., 2016; Han et al., 2017; Thanh Nguyen et al., 2016; Yang et al., 2018; Wang et al., 2017). As quantizing shapes to voxel grids lead to geometric information loss, recent approaches (Yuan et al., 2018; Yu et al., 2018b; Achlioptas et al., 2018) operate directly on point sets to fill in missing parts. These works, however, require supervision in the form of partial-complete paired data for training deep neural networks to directly regress partial input to their ground truth counterparts. Since paired ground truth of real-world data is rarely available such training data is generated using virtual scanning. While the methods work well on synthetic test data, they do not generalize easily to real scans arising from hard-to-model acquisition processes.

Realizing the gap between synthetically-generated data and real-world data, Stutz & Geiger (2018) proposed to directly work on voxelized real-world data. They also work in a latent space created for clean and complete data but measure reconstruction loss using a maximum likelihood estimator. Instead, we propose a GAN setup to learn a mapping between latent spaces respectively arising from partial real and synthetic complete data. Further, by measuring loss using Hausdorff distance on point clouds, we directly work with point sets instead of voxelized input.

**Generative Adversarial Network.**     Since its introduction, GAN (Goodfellow et al., 2014) has been used for a variety of generative tasks. In 2D image domain, researchers have utilized adversarial training to recover richer information from low-resolution images or corrupted images (Ledig et al., 2017; Wang et al., 2018; Mao et al., 2017; Park et al., 2018; Bulat et al., 2018; Yeh et al., 2017; Iizuka et al., 2017). In 3D context, Yang et al. (2018); Wang et al. (2017) combine 3D-CNN and generative adversarial training to complete shapes under the supervision of ground truth data. Gurumurthy & Agrawal (2019) treats the point cloud completion task as denoising AE problem, utilizing adversarial training to optimize on the AE latent space. We also leverage the power of GAN for reasoning the missing part of partial point cloud scanning. However, our method is designed to work with unpaired data, and thus can directly be applied to real-world scans even when real-world and synthetic data distributions differ. Intuitively, our GAN-based approach directly learns a translation mapping between these two different distributions.

**Deep Learning on Point clouds.**     Our method is built upon recent advances in deep neural networks for point clouds. PointNet Qi et al. (2017a), the pioneering work on this topic, takes an input point set through point-wise MLP layers followed by a symmetric and permutation-invariant func-

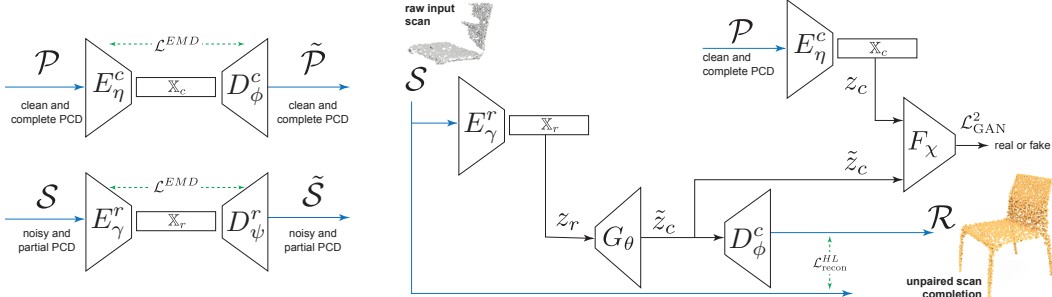

Figure 2: Unpaired Scan Completion Network.

tion to produce a compact global feature, which can then be used for a diverse set of tasks (e.g., classification, segmentation). Although many improvements to PointNet have been proposed (Su et al., 2018; Li et al., 2018b; Qi et al., 2017b; Li et al., 2018a; Zaheer et al., 2017), the simplicity and effectiveness of PointNet and its extension PointNet++ make them popular for many other analysis tasks (Yu et al., 2018a; Yin et al., 2018; Yu et al., 2018b; Guerrero et al., 2018).

In the context of synthesis, Achlioptas et al. (2018) proposed an autoencoder network, using a PointNet-based backbone, to learn compact representations of point clouds. By working in a reduced latent space produced by the autoencoder, they report significant advantages in training GANs, instead of having a generator producing raw point clouds. Inspired by this work, we design a GAN to translate between two different latent spaces to perform unpaired shape completion on real scans.

## 3 METHOD

Given a noisy and partial point set $\mathcal{S} = \{\mathbf{s}_i\}$ as input, our goal is to produce a clean and complete point set $\mathcal{R} = \{\mathbf{r}_i\}$ as output. Note that although the two sets have the same number of points, there is no explicit correspondence between the sets $\mathcal{S}$ and $\mathcal{R}$. Further, we assume access to clean and complete point sets for shapes for the object classes. We achieve unpaired completion by learning two class-specific point set manifolds, $\mathbb{X}_r$ for the scanned inputs, and $\mathbb{X}_c$ for clean and complete shapes. Solving the shape completion problem then amounts to learning a mapping $\mathbb{X}_r \to \mathbb{X}_c$ between the respective latent spaces. We train a generator $G_\theta : \mathbb{X}_r \to \mathbb{X}_c$ to perform the mapping. Note that we do *not* require the noise characteristics in the two data distributions, i.e., real and synthetic, to be the same. In absence of paired training data, we score the generated output by setting up a min-max game where the generator is trained to fool a discriminator $F_\chi$, whose goal is to differentiate between encoded clean and complete shapes, and mapped encodings of the raw and partial inputs. Figure 2 shows the setup of the proposed scan completion network. The latent space encoder-decoders, the mapping generator, and the discriminator are all trained as detailed next.

### 3.1 LEARNING LATENT SPACES FOR POINT SETS

The latent space of a given set of point sets is obtained by training an autoencoder, which encodes the given input to a low-dimension latent feature and then decodes to reconstruct the original input. We work directly on the point sets via these learned latent spaces instead of quantizing them to voxel grids or signed distance fields.

For point sets coming from the clean and complete point sets $\mathcal{P}$, we learn an encoder network $E_\eta^c$ that maps $\mathcal{P}$ from the original parameter space $\mathbb{R}^{3N}$, defined by concatenating the coordinates of the $N$ (2048 in all our experiments) points, to a lower-dimensional latent space $\mathbb{X}_c$. A decoder network $D_\phi^c$ performs the inverse transformation back to $\mathbb{R}^{3N}$ giving us a reconstructed point set $\tilde{\mathcal{P}}$ with also $N$ points. The encoder-decoders are trained with reconstruction loss,

$$\mathcal{L}^{\text{EMD}}(\eta, \phi) = \mathbb{E}_{\mathcal{P} \sim p_{\text{complete}}} d(\mathcal{P}, D_\phi^c(E_\eta^c(\mathcal{P}))), \tag{1}$$

where $\mathcal{P} \sim p_{\text{complete}}$ denotes point set samples drawn from the set of clean and complete point sets, $d(X_1, X_2)$ is the Earth Mover's Distance (EMD) between point sets $X_1, X_2$, and $(\eta, \phi)$ are the

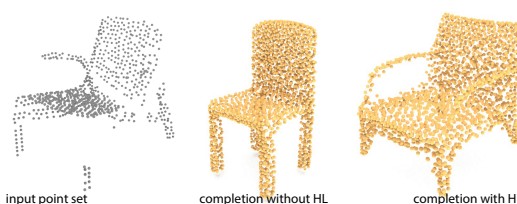

input point set    completion without HL    completion with HL

Figure 3: Effect of unpaired scan completion without (Equation 5) and with HL term (Equation 6). Without the HL term, the network produces a clean point set for a complete chair, that is different in shape from the input. With the HL term, the network produces a clean point set that matches the input.

learnable parameters of the encoder and decoder networks, respectively. Once trained, the weights of both networks are held fixed and the latent code $z = E^c_\eta(X)$, $z \in \mathbb{X}_c$ for a clean and complete point set $X$ provides a compact representation for subsequent training and implicitly captures the manifold of clean and complete data. The architecture of the encoder and decoder is similar to Achlioptas et al. (2018); Qi et al. (2017a): using a 5-layer MLP to lift individual points to a deeper feature space, followed by a symmetric function to maintain permutation invariance. This results in a $k$-dimensional latent code that describes the entire point cloud ($k$=128 in all our experiments). More details of the network architecture can be found in the appendix.

As for the point set coming from the noisy-partial point sets $\mathcal{S}$, one can also train another encoder $E^r_\gamma : \mathcal{S} \to \mathbb{X}_r$ and decoder $D^r_\psi : \mathbb{X}_r \to \tilde{\mathcal{S}}$ pair that provides a latent parameterization $\mathbb{X}_r$ for the noisy-partial point sets, with the definition of the reconstruction loss as,

$$\mathcal{L}^{\text{EMD}}(\gamma, \psi) = \mathbb{E}_{\mathcal{S} \sim p_{\text{raw}}} d(\mathcal{S}, D^r_\psi(E^r_\gamma(\mathcal{S}))), \tag{2}$$

where $\mathcal{S} \sim p_{\text{raw}}$ denotes point set samples drawn from the set of noisy and partial point sets.

Although, in experiments, the latent space of this autoencoder trained on noisy-partial point sets works considerably well as the noisy-partial point set manifold, we found that using the latent space produced by feeding noisy-partial point sets to the autoencoder trained on clean and complete point sets yields slightly better results. Hence, unless specified, we set $\gamma = \eta$ and $\psi = \phi$ in our experiments. The comparison of different choices to obtain the latent space for noisy-partial point sets is also presented in Section 4. Next, we will describe the GAN setup to learn a mapping between the latent spaces of raw noisy-partial and synthetic clean-complete point sets, i.e., $\mathbb{X}_r \to \mathbb{X}_c$.

### 3.2 LEARNING A MAPPING BETWEEN LATENT SPACES

We set up a min-max game between a generator and a discriminator to perform the mapping between the latent spaces. The generator $G_\theta$ is trained to perform the mapping $\mathbb{X}_r \to \mathbb{X}_c$ such that the discriminator fails to reliably tell if the latent variable comes from original $\mathbb{X}_c$ or the remapped $\mathbb{X}_r$.

The latent representation of a noisy and partial scan $z_r = E^r_\gamma(\mathcal{S})$ is mapped by the generator to $\tilde{z}_c = G_\theta(z_r)$. Then, the task of the discriminator $F_\chi$ is to distinguish between latent representations $\tilde{z}_c$ and $z_c = E^c_\eta(\mathcal{P})$. We train the mapping function using a GAN. Given training examples of clean latent variables $z_c$ and remapped-noisy latent variables $\tilde{z}_c$, we seek to optimize the following adversarial loss over the mapping generator $G_\theta$ and a discriminator $F_\chi$,

$$\min_\theta \max_\chi \mathbb{E}_{x \sim p_{\text{clean-complete}}} \left[ \log \left( F_\chi \left( E^c_\eta(x) \right) \right) \right] + \mathbb{E}_{y \sim p_{\text{noisy-partial}}} \left[ \log \left( 1 - F_\chi \left( G_\theta(E^r_\gamma(y)) \right) \right) \right]. \tag{3}$$

In our experiments, we found the least square GAN Mao et al. (2016) to be easier to train and hence minimize both the discriminator and generator losses defined as,

$$\mathcal{L}_F(\chi) = \mathbb{E}_{x \sim p_{\text{clean-complete}}} \left[ F_\chi \left( E^c_\eta(x) \right) - 1 \right]^2 + \mathbb{E}_{y \sim p_{\text{noisy-partial}}} \left[ F_\chi \left( G_\theta(E^r_\gamma(y)) \right) \right]^2 \tag{4}$$

$$\mathcal{L}_G(\theta) = \mathbb{E}_{y \sim p_{\text{noisy-partial}}} \left[ F_\chi \left( G_\theta(E^r_\gamma(y)) \right) - 1 \right]^2. \tag{5}$$

The above setup encourages the generator to perform the mapping $\mathbb{X}_r \to \mathbb{X}_c$ resulting in $D^c_\psi(\tilde{z}_c)$ to be a clean and complete point cloud $\mathcal{R}$. However, the generator is free to map a noisy latent vector to any point on the manifold of valid shapes in $\mathbb{X}_c$, including shapes that are far from the original partial scan $\mathcal{S}$. As shown in Figure 3, the result is a complete and clean point cloud that can be dissimilar in shape to the partial scanned input. To prevent this, we add a reconstruction loss term $\mathcal{L}_{\text{recon}}$ to the generator loss:

$$\mathcal{L}_G(\theta) = \alpha \mathbb{E}_{y \sim p_{\text{noisy-partial}}} \left[ F_\chi \left( G_\theta(E^r_\gamma(y)) \right) - 1 \right]^2 + \beta \mathcal{L}^{\text{HL}}_{\text{recon}}(\mathcal{S}, D^c_\psi(G_\theta(E^r_\gamma(\mathcal{S})))), \tag{6}$$

where $\mathcal{L}_{\mathrm{recon}}^{\mathrm{HL}}$ denotes the Hausdorff distance loss [1] (HL) from the partial input point set to the completion point set, which encourages the predicted completion point set to match the input only *partially*. Note that, it is crucial to use HL as $\mathcal{L}_{\mathrm{recon}}$, since the partial input can only provide *partial* supervision when no ground truth complete point set is available. In contrast, using EMD as $\mathcal{L}_{\mathrm{recon}}$ forces the network to reconstruct the overall partial input leading to worse completion results. The comparison of these design choices is presented in Section 4. Unless specified, we set the trade-off parameters as $\alpha = 0.25$ and $\beta = 0.75$ in all our experiments.

## 4 EXPERIMENTAL EVALUATION

We present quantitative and qualitative experimental results on several noisy and partial datasets. First, we present results on real-world datasets, demonstrating the effectiveness of our method on unpaired raw scans. Second, we thoroughly compare our method to various baseline methods on 3D-EPN dataset, which contains simulated partial scans and corresponding ground truth for full evaluation. Finally, we derive a synthetic noisy-partial scan dataset based on ShapeNet, on which we can evaluate the performance degradation of applying supervised methods to test data of different distribution and the performance of our method under varying levels of incompleteness. A set of ablation studies is also included to evaluate our design choices.

**Datasets.** *(A) Real-world dataset* comes from three sources. First, a dataset of $\sim$550 chairs and $\sim$550 tables extracted from the ScanNet dataset split into 90%-10% train-test sets. Second, a dataset of 20 chairs and 20 tables extracted from the Matterport3D dataset. Note that we train our method only on the ScanNet training split, and use the trained model to test on the Matterport3D data to evaluate generalization to new data sources. Third, a dataset containing cars from the KITTI Velodyne point clouds. *(B) 3D-EPN dataset* provides simulated partial scans with corresponding ground truth. Scans are represented as Signed Distance Field (SDF). We only use the provided point cloud representations of the training data, instead of using the SDF data which holds richer information. *(C) Clean and complete point set dataset* contains virtually scanned point sets of ShapeNet models covering 8 categories, namely boat, car, chair, dresser, lamp, plane, sofa, and table. We use this dataset for learning the clean-complete point set manifold in all our experiments. *(D) Synthetic dataset* provides different incomplete scan distribution and at different levels of incompleteness. Ground truth complete scan counterparts are available for evaluation.

**Evaluation measures.** We assess completion quality using the following measures. *(A) Accuracy* measures the fraction of points in $P_{comp}$ that are matched by $P_{gt}$, where and $P_{comp}$ denote the completion point set and $P_{gt}$ denote the ground truth point set. Specifically, for each point $v \in P_{comp}$, we compute $D(v, P_{gt}) = min\{\| v - q \|, q \in P_{gt}\}$. If $D(v, P_{gt})$ is within distance threshold $\epsilon = 0.03$, we count it as a correct match. The fraction of matched points is reported as the accuracy in *percentage*. *(B) Completeness* reports the fraction of points in $P_{gt}$ that are within distance threshold $\epsilon$ of any point in $P_{comp}$. *(C) F1 score* is defined as the harmonic average of the accuracy and the completeness, where F1 reaches its best value at 1 (perfect accuracy and completeness) and worst at 0. *(D) Plausibility* of the completion is evaluated as the classification accuracy in *percentage* produced by PointNet++, a SOA point-based classification network. To avoid bias on ShapeNet point clouds, we trained the classification network on the ModelNet40 dataset. We mainly used plausibility score for real-world data completions, where no ground truth data is available for calculating accuracy, completeness, or F1 scores.

In the following, we show all experimental and evaluation results. We trained separate networks for each category. More details are in the appendix.

### 4.1 EXPERIMENTAL RESULTS ON REAL-WORLD DATA

Our method works directly on real-world data where no paired data is available. We train and test our network on noisy-partial chairs and tables extracted from the ScanNet dataset. We further test the network trained on ScanNet dataset on chairs and tables extracted from the Matterport3D dataset, to show how well our network can generalize to definitely unseen data. We present qualitative results of our method in Fig 4. Our method consistently produces plausible completions for the ScanNet and Matterport3D data.

---

[1]Unless specified, we use unidirectional Hausdorff distance, from the partial input to its completion.

Figure 4: Qualitative comparisons on real-world data, which includes partial scans of ScanNet chairs and tables, Matterport3D chairs and tables, and KITTI cars. We show the partial input in grey and the corresponding completion in gold on the right.

In the absence of ground truth completions on real data, we compare our method quantitatively against others based on the plausibility of the results. The left sub-table of Table 1 shows that our method is superior to those supervised methods, namely 3D-EPN and PCN. Directly applying PCN trained on simulated partial data to real-world data leads to completions that have low plausibility, while our method consistently produces results with high plausibility. 3D-EPN trained on simulated partial data failed to complete the real-world partial scans. In Section 4.2 and Section 4.3, we present more in-depth comparisons on 3D-EPN and our synthetic dataset, where the ground truth is available for computing accuracy, completeness, and F1 of the completions.

Table 1: **Completion plausibility on synthetic scans and real-world scans and effects of data distribution discrepancy.** (Left) Plausibility comparison on synthetic scans and real-world scans. Synthetic scans includes test data from 3D-EPN, real-world scans includes ScanNet and Matterport3D test data. 3D-EPN failed to produce good completions on real-world data. (Right) On our synthetic data, supervised methods trained on other simulated partial scans produce worse results on partial scans with different data distribution.

|  |  | Raw input | 3D-EPN | PCN | Ours |
|---|---|---|---|---|---|
| Synthetic | chair | 73.1 | 77.3 | 85.0 | **91.5** |
|  | table | 52.5 | 71.2 | 72.0 | **80.6** |
| Real-world | chair | 71.4 | 7.1 | 78.6 | **94.3** |
|  | table | 47.8 | 4.4 | 69.6 | **81.2** |

|  | 3D-EPN | | | PCN | | | Ours | | |
|---|---|---|---|---|---|---|---|---|---|
| model | acc. | comp. | F1 | acc. | comp. | F1 | acc. | comp. | F1 |
| chair | 39.6 | 61.8 | 48.2 | 49.3 | 76.0 | 59.8 | 80.7 | 80.8 | **80.8** |
| car | 43.8 | 62.3 | 51.4 | 63.2 | 81.4 | 71.2 | 82.6 | 80.7 | **81.7** |
| table | 36.6 | 61.0 | 45.8 | 62.3 | 80.6 | 70.3 | 83.1 | 84.5 | **83.8** |
| plane | 17.1 | 57.6 | 26.3 | 67.1 | 85.4 | 75.1 | 94.4 | 92.7 | **93.6** |

Completing the car observations from KITTI is extremely challenging, as each car instance only receives few data points from the Lidar scanner. Fig 4 shows the qualitative results of our method on completing sparse point sets of KITTI cars, we can see that our network can still generate highly plausible cars with such sparse inputs.

We also use a point-based object part segmentation network (Qi et al., 2017b) to indirectly evaluate our completions of real-world data. Due to the absence of ground truth segmentation, we calculate the approximate segmentation accuracy for each completion. Specifically, for the completion of a chair, we count the predicted segmentation label of each point to be correct as long as the predicted label falls into the set of 4 parts (i.e., seat, back, leg, and armrest) of chair class. Our completion results have much higher approximate segmentation accuracy compared to the real-world raw input (chair: 77.2% *vs.* 24.8%; table: 96.4% *vs.* 83.5%; and car: 98.0% *vs.* 5.2%, as segmentation accuracy on our completions versus on original partial input), indicating high completion quality.

## 4.2 COMPARISON WITH BASELINES ON 3D-EPN DATA

We compare our method to several baseline methods and present both quantitative and qualitative comparisons on the 3D-EPN test set:

- Autoencoder (AE), which is trained only with clean and complete point sets.
- 3D-EPN (Dai et al., 2017b), a supervised method that requires SDF input and is trained with paired data. We convert its Distance Field representation results into surface meshes, from which we can uniformly sample $N$ points for calculating our point-based measures.

Table 2: **Comparison with baselines on the 3D-EPN dataset**. Note that 3D-EPN and PCN require paired supervision data, while ours does not. Ours outperforms 3D-EPN and achieves comparable results to PCN. Furthermore, after adapted to leverage the ground truth data as well, our method achieves similar performance to PCN.

| | AE | | | EPN (**fully supervised**) | | | PCN (**fully supervised**) | | | Ours (**unsupervised**) | | | Ours+ (supervised) | | |
|---|---|---|---|---|---|---|---|---|---|---|---|---|---|---|---|
| model | acc. | comp. | F1 | acc. | comp. | F1 | acc. | comp. | F1 | acc. | comp. | F1 | acc. | comp. | F1 |
| boat | 89.6 | 81.4 | 85.3 | 82.4 | 81.4 | 81.9 | 92.6 | 93.4 | **93.0** | 86.6 | 84.7 | 85.6 | 89.8 | 92.0 | 90.9 |
| car | 81.3 | 71.1 | 75.9 | 69.8 | 81.7 | 75.3 | 97.3 | 96.1 | **96.7** | 88.9 | 87.6 | 88.2 | 93.5 | 92.8 | 93.1 |
| chair | 79.9 | 68.5 | 73.8 | 61.7 | 76.9 | 68.5 | 91.1 | 90.6 | **90.9** | 78.7 | 77.4 | 78.0 | 82.3 | 83.3 | 82.8 |
| dresser | 68.9 | 64.2 | 66.5 | 58.4 | 72.7 | 64.8 | 93.5 | 91.5 | **92.5** | 75.8 | 76.5 | 76.2 | 87.4 | 91.5 | 89.4 |
| lamp | 75.9 | 79.6 | 77.7 | 60.8 | 67.8 | 64.1 | 82.9 | 88.3 | **85.5** | 71.3 | 80.2 | 75.5 | 76.6 | 86.3 | 81.2 |
| plane | 97.6 | 95.1 | 96.3 | 78.1 | 93.5 | 85.1 | 98.3 | 98.2 | **98.2** | 97.2 | 95.9 | 96.5 | 95.6 | 94.8 | 95.2 |
| sofa | 80.3 | 64.0 | 71.2 | 65.0 | 72.6 | 68.6 | 91.5 | 90.8 | **91.1** | 68.2 | 72.3 | 70.2 | 81.0 | 87.0 | 83.9 |
| table | 82.8 | 72.5 | 77.3 | 56.8 | 75.1 | 64.7 | 93.4 | 89.2 | **91.2** | 82.2 | 77.8 | 80.0 | 81.2 | 81.4 | 81.3 |

- PCN (Yuan et al., 2018), which completes partial inputs in a hierarchical manner, receiving supervision from both sparse and dense ground truth point clouds.

- Ours+, which is an adaption of our method for training with paired data, to show that our method can be easily adapted to work with ground truth data, improving the completion. Specifically, we set $\alpha = 0$ and use EMD loss as $L_{recon}$. More details and discussion about adapting our method to train with paired data can be found in the appendix.

Table 2 shows quantitative results on 3D-EPN test split and summarizes the comparisons: although our network is only trained with unpaired data, our method outperforms 3D-EPN method and achieves comparable results to PCN. Note that both 3D-EPN and PCN require paired data. Furthermore, after adapting our method to be supervised by the ground truth, the performance of our method (Ours+) improves, achieving similar performance to PCN. Note that a simple autoencoder network trained with only clean-complete data can produce quantitatively good results, especially when the input is rather complete. Thus, we also evaluate the performance of AE on our synthetic data with incompleteness control in Section 4.3, to show that AE performance declines dramatically as the incompleteness of the input increases. Additional comparisons are included in the appendix.

### 4.3 EFFECT OF DATA DISTRIBUTION DISCREPANCY AND VARYING INCOMPLETENESS

Supervised methods assume that simulated partial scans share the same data distribution as the test data. We conduct quantitative experiments to show that applying 3D-EPN and PCN to our synthetic data, which is of different data distribution to its training data and in which the ground truth complete scans are not available for training, lead to performance degradation. The right sub-table of Table 1 shows that our method continues to produce good completions on our synthetic data, as we do not require paired data for training. The visual comparison is presented in the appendix.

To evaluate our method under different levels of input incompleteness, we conduct experiments on our synthetic data, in which we can control the fraction of missing points. Specifically, we train our network with varying levels of incompleteness by randomizing the amount of missing points during training, and afterwards fix the amount of missing points for testing. Table 3 shows the performance of our method on different classes under increasing amount of incompleteness and the comparison to AE. We can see that AE performance declines dramatically as the incompleteness of the input increases, while our method can still produce completions with high plausibility and F1 score.

Table 3: **Effect of varying incompleteness.** Performance of AE and ours with increasing incompleteness (% of the missing points). Our completions remain robust even with increasing incompleteness as our method restricts the completion via the learned latent shape manifolds.

| | | Plausibility | | F1 | | | Plausibility | | F1 | | | Plausibility | | F1 | | | Plausibility | | F1 | |
|---|---|---|---|---|---|---|---|---|---|---|---|---|---|---|---|---|---|---|---|---|---|
| incomp. | model | AE | Ours | AE | Ours | model | AE | Ours | AE | Ours | model | AE | Ours | AE | Ours | model | AE | Ours | AE | Ours |
| 10 | | 96.3 | **99.4** | **94.9** | 85.8 | | 88.0 | **91.0** | **88.4** | 87.7 | | 69.3 | **74.0** | 90.0 | **90.2** | | 88.9 | **91.0** | **96.6** | 96.5 |
| 20 | | 96.7 | **99.7** | **89.5** | 84.5 | | 81.0 | **91.0** | **87.2** | 85.1 | | 65.0 | **75.5** | 85.0 | **87.3** | | 89.7 | **90.7** | 94.0 | **95.5** |
| 30 | car | 95.0 | **98.2** | 81.8 | **83.3** | chair | 67.0 | **90.9** | 70.8 | **80.7** | table | 55.2 | **73.4** | 77.3 | **84.0** | plane | 88.7 | **90.7** | 89.5 | **94.1** |
| 40 | | 85.4 | **96.1** | 71.8 | **79.7** | | 44.0 | **89.4** | 52.5 | **76.9** | | 45.1 | **71.8** | 69.6 | **80.1** | | 85.0 | **89.0** | 84.9 | **92.8** |
| 50 | | 58.6 | **96.4** | 63.4 | **72.5** | | 38.0 | **83.5** | 33.5 | **71.8** | | 32.5 | **73.3** | 62.1 | **74.5** | | 80.0 | **90.7** | 80.6 | **91.0** |

### 4.4 DIVERSITY OF THE COMPLETION RESULTS

Our network is encouraged to partially match the input, alleviating the mode-collapse issue, which often occurs in GAN. Unlike traditional generative model problem, where high diversity in generated results is always better, the diversity in our completion results should match that of the ground truth. Although this can be qualitatively accessed, see Fig. 5 in Appendix, in order to quantitatively quantify the divergence, we compute the Jensen-Shannon Divergence (JSD) between the marginal distribution of ground truth point sets and that of our completions as proposed in Achlioptas et al. (2018). As a reference, we simulate extremely mode-collapsed point cloud sets by repeating a randomly selected point cloud, then report the JSD between ground truth point sets and the simulated extremely mode-collapsed point sets. The JSD scores – *lower is better* – highlight the diversity of our 3D-EPN completions and the divergence between our diversity and that of the ground truth using the extreme mode-collapse results as reference (the former is ours): 0.06 *vs.* 0.46 on cars, 0.05 *vs.* 0.61 on chairs, 0.04 *vs.* 0.53 on planes and 0.04 *vs.* 0.59 on tables.

### 4.5 ABLATION STUDY

- Ours with partial AE, uses encoder $E_\gamma^r$ and decoder $D_\psi^r$ that are trained to reconstruct partial point sets for the latent space of partial input.

- Ours with EMD loss, uses EMD as the reconstruction loss.

- Ours without GAN, "switch off" the GAN module by simply setting $\alpha = 0$ and $\beta = 1$, to verify the effectiveness of using adversarial training in our network.

- Ours with reconstruction loss, removes the reconstruction loss term by simply setting $\alpha = 1$ and $\beta = 0$, to verify the effectiveness of the reconstruction loss term in generator loss.

Table 4 presents quantitative results for the ablation experiments, where we demonstrate the importance of various design choices and modules in our proposed network. We can see that our method has the best performance over all other variations.

Table 4: Ablation study showing the importance of various design choices in our proposed network.

| | Ours w/ partial AE | | | Ours w/ EMD | | | Ours w/o GAN | | | Ours w/o Recon. | | | Ours | | |
|---|---|---|---|---|---|---|---|---|---|---|---|---|---|---|---|
| | acc. | comp. | F1 | acc. | comp. | F1 | acc. | comp. | F1 | acc. | comp. | F1 | acc. | comp. | F1 |
| boat | 75.1 | 75.4 | 75.2 | 82.0 | 84.8 | 83.4 | 47.4 | 93.1 | 62.8 | 44.4 | 38.1 | 41.0 | 86.6 | 84.7 | **85.6** |
| car | 88.9 | 87.6 | 88.2 | 76.0 | 76.8 | 76.4 | 46.2 | 88.3 | 60.7 | 72.2 | 72.7 | 72.5 | 88.9 | 87.7 | **88.3** |
| chair | 64.1 | 66.7 | 65.4 | 78.6 | 76.4 | 77.5 | 41.3 | 79.8 | 54.4 | 75.6 | 75.1 | 75.3 | 78.7 | 77.4 | **78.0** |
| dresser | 67.4 | 68.6 | 68.0 | 71.4 | 72.3 | 71.9 | 44.2 | 74.4 | 55.4 | 20.9 | 21.9 | 21.4 | 75.8 | 76.5 | **76.2** |
| lamp | 64.0 | 74.8 | 69.0 | 69.9 | 79.0 | 74.2 | 28.6 | 84.7 | 42.8 | 15.6 | 22.2 | 18.3 | 71.3 | 80.2 | **75.5** |
| plane | 94.3 | 94.9 | 94.6 | 96.8 | 95.4 | 96.1 | 41.2 | 98.3 | 58.1 | 87.1 | 84.7 | 85.9 | 97.2 | 95.9 | **96.5** |
| sofa | 64.8 | 67.3 | 66.0 | 68.6 | 69.8 | 69.2 | 38.6 | 75.6 | 51.1 | 55.1 | 58.0 | 56.5 | 68.2 | 72.3 | **70.2** |
| table | 76.0 | 77.6 | 76.8 | 81.5 | 75.1 | 78.2 | 23.0 | 59.3 | 33.1 | 27.4 | 23.4 | 25.2 | 82.2 | 77.8 | **80.0** |

## 5 CONCLUSION

We presented a point-based unpaired shape completion framework that can be applied directly on raw partial scans to obtain clean and complete point clouds. At the core of the algorithm is an adaptation network acting as a generator that transforms latent code encodings of the raw point scans, and maps them to latent code encodings of clean and complete object scans. The two latent spaces regularize the problem by restricting the transfer problem to respective data manifolds. We extensively evaluated our method on real scans and virtual scans, demonstrating that our approach consistently leads to plausible completions and perform superior to other methods. The work opens up the possibility of generalizing our approach to scene-level scan completions, rather than object-specific completions. Our method shares the same limitations as many of the supervised counterparts: does not produce fine-scale details and assumes input to be canonically oriented. Another interesting future direction will be to combine point- and image-features to apply the completion setup to both geometry and texture details.

## 6 ACKNOWLEDGEMENTS

We thank all the anonymous reviewers for their insightful comments and feedback. This work is supported in part by grants from National Key R&D Program of China (2019YFF0302900), China Scholarship Council, National Natural Science Foundation of China (No.61602273), ERC Starting Grant, ERC PoC Grant, Google Faculty Award, Royal Society Advanced Newton Fellowship, and gifts from Adobe.

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

## A    DETAILS OF DATASETS

**Clean and Complete Point Sets** are obtained by virtually scanning the models from ShapeNet. We use a subset of 8 categories, namely boat, car, chair, dresser, lamp, plane, sofa and table, in our experiments. To generate clean and complete point set of a model, we virtually scan the models by performing ray-intersection test from cameras placed around the model to obtain the dense point set, followed by a down-sampling procedure to obtain a relatively sparser point set of $N$ points. Note that we use the models without any pose and scale augmentation.

This dataset is used for training to learn the clean-complete point set manifold in all our experiments. The following datasets of different data distributions serve as different noisy-partial input data.

**Real-world Data** comes from three sources. The first one is derived from ScanNet dataset which provides many mesh objects that have been pre-segmented from its surrounding environment. For the purpose of training and testing our network, we extract ∼550 chair objects and ∼550 table objects from ScanNet dataset, and manually align them to be consistently orientated with models in ShapeNet dataset. We also split these objects into 90%/10% train/test sets.

The second one consists of 20 chairs and 20 tables from the Matterport3D dataset, to which the same extraction and alignment as is done in ScanNet dataset is also applied. Note that we train our method only on ScanNet training split, and use the trained model to test on Matterport3D data, to show how our method can generalize to absolutely unseen data. For both ScanNet and Matterport3D datasets, we uniformly sample $N$ points on the surface mesh of each object to obtain the input point sets.

Last, we extract car observations from the KITTI dataset using the provided ground truth bounding boxes for training and testing our method. We use KITTI Velodyne point clouds from the 3D object detection benchmark and the split of Qi et al. (2018). We filter the observations such that each car observation contains at least 100 points to avoid overly sparse observations.

**3D-EPN Dataset** provides partial reconstructions of ShapeNet objects (8 categories) by using volumetric fusion method Curless & Levoy (1996) to integrate depth maps scanned along a virtual scanning trajectory around the model. For each model, a set of trajectories is generated with different levels of incompleteness, reflect the real-world scanning with a hand-held commodity RGB-D sensor. The entire dataset covers 8 categories and a total of 25590 object instances (the test set is composed of 5384 models). Note that, in the original 3D-EPN dataset, the data is represented in Signed Distance Field (SDF) for training data and Distance Field (DF) for test data. As our method works on pure point sets, we only use the point cloud representations of the training data provided by the authors, instead of using the SDF data which holds richer information and is claimed in Dai et al. (2017b) to be crucial for completing partial data.

**Synthetic Data** serves the purpose of having another dataset of different incomplete scan distribution and controlling the incompleteness of the input. We use ShapeNet to generate a synthetic dataset, in which we can control the incompleteness of the synthetic partial point sets. For the models in each one of the 4 categories (car, chair, plane, and table), we split them into 90%/10% train/test sets. For each model, from which a clean and complete point set has been scanned (as described earlier in this subsection), we can randomly pick a point and remove its $N \times r$ ($r \in [0, 1)$) nearest neighbor points. The parameter $r$ controls the incompleteness of the synthetically-generated input. Furthermore, we add Gaussian noise $\mathcal{N}(\mu, \sigma^2)$ to each point ($\mu$=0 and $\sigma$=0.01 for all our experiments). Last, we duplicate the points in the resulting point sets to generate point sets with an equal number of $N$ points.

## B    NETWORK ARCHITECTURE DETAILS

In this section, we describe the details of the encoder, decoder, generator and discriminator in our network implementation.

### B.1    AE ARCHITECTURE DETAILS

**Encoder**    consists of 5 1-D convolutional layers which are implemented as 1-D convolutions with ReLU and batch normalization, with kernel size of 1 and stride of 1, to lift the feature of each point to high dimensional feature space independently. In all experiments, we use an encoder with 64,

128, 128, 256 and $k = 128$ filters in each of its layers, with $k$ being the latent code size. The output of the last convolutional layer is passed to a feature-wise maximum to produce a $k$-dimensional latent code.

**Decoder** transforms the latent vector using 3 fully connected layers with 256, 256, and $N$ x 3 neurons each, the first two having ReLUs, to reconstruct $N$ x 3 output.

## B.2 GAN ARCHITECTURE DETAILS

Since the generator and discriminator of GAN directly operate on the latent space, the architecture for them is significantly simpler. Specifically, the generator is comprised of two fully connected layers with 128 and 128 neurons each, to map the latent code of noisy and incomplete point sets to that of clean and complete point sets. The discriminator consists of 3 fully connected layers with 256, 512 and 1 neurons each, to produce a single scalar for each latent code.

## C TRAINING DETAILS

To make the training of the entire network trackable, we pre-train the AEs used for obtaining the latent spaces. After that we retain the weights of AEs, only the weights of the generator and discriminator are updated through the back-propagation during the GAN training. The following training hyper-parameters are used in all our experiments.

For training the AE, we use Adam optimizer with an initial learning rate of 0.0005, $\beta_1 = 0.9$ and a batch size of 200 and train for a maximum of 2000 epochs.

For training the generator and discriminator on the latent spaces, we use Adam optimizer with an initial learning rate of 0.0001, $\beta_1 = 0.5$ and a batch size of 24 and train the generator and discriminator alternately for a maximum of 1000 epochs.

## D QUALITATIVE RESULTS ON 3D-EPN DATASET

We present qualitatively comparisons in Fig 5, where we show the partial input, AE, 3D-EPN, PCN, Ours, Ours+ result and the ground truth point set. We can see that, although our method is not quantitatively the best, our results are very qualitatively plausible, as the generator is restricted to generate point sets from learned clean and complete shape manifolds.

## E VISUAL COMPARISON ON TEST DATA WITH DISTRIBUTION DIFFERENT TO TRAINING DATA

We present the visual comparison of 3D-EPN, PCN and our method on our synthetic data, which differs from 3D-EPN and PCN training data. In this experiment, the ground truth of our synthetic data is only available for evaluation. In Fig 6, we can see that our method keeps producing high-quality completions, as our method does not require paired data for training hence can still be trained when no ground truth is available. 3D-EPN and PCN produce much worse results as the data distribution of its training data and our synthetic data differ.

## F VARIATIONS OF LEVERAGING GROUND TRUTH SUPERVISION

To adapt our network for training with ground truth point sets, we first change the reconstruction loss term in the generator loss from HD (Hausdorff Distance) to EMD (Earth Mover's Distance), as the ground truth point set is complete and thus contains full information for supervising the completion. Note that HD is superior when the ground truth is unavailable for training, as shown in Table 4 of Section 4. Moreover, we present the comparison of different decisions on whether to adopt the adversarial training in our network for training with the ground truth.

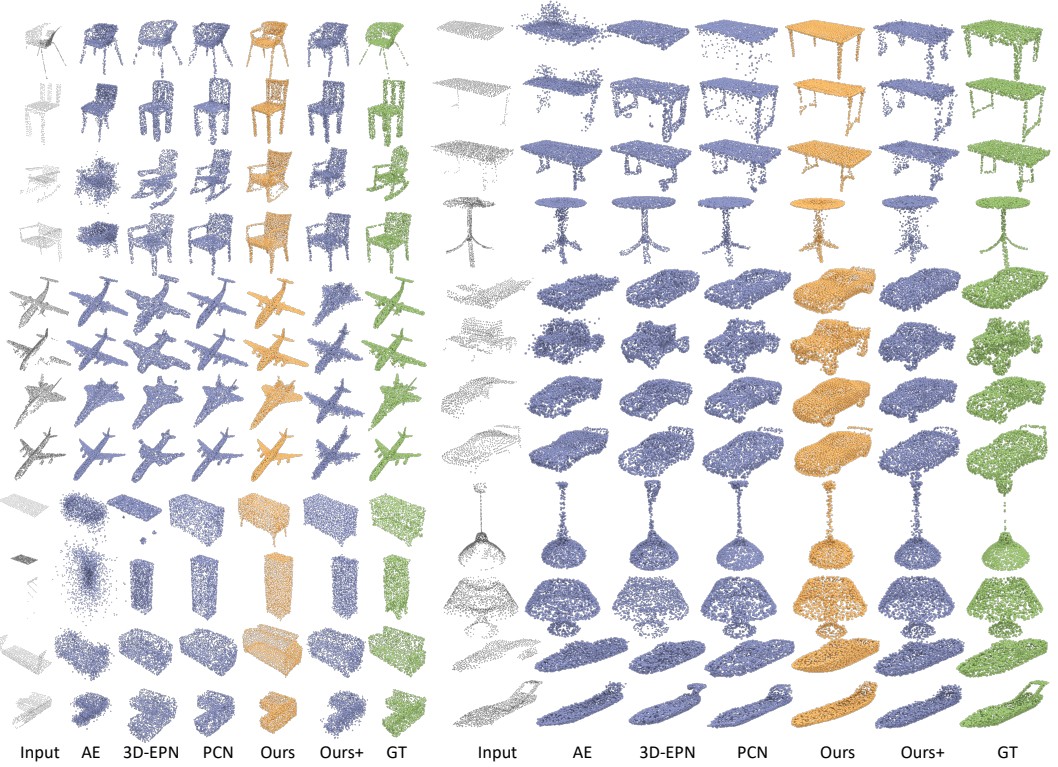

Figure 5: Qualitative comparison on 3D-EPN dataset.

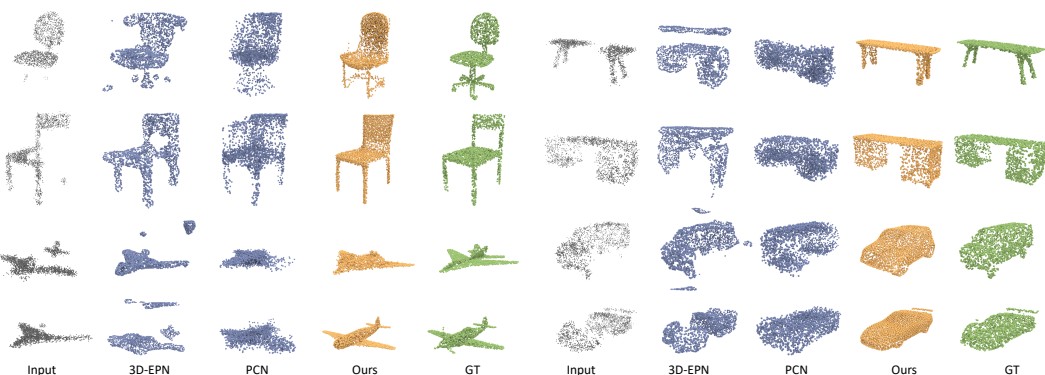

Figure 6: Effect of data distribution discrepancy and qualitative comparison on our synthetic dataset.

- Ours (GT+EMD), which is also denoted as Ours+ in the paper, removes the adversarial training in the network by simply setting $\alpha = 0$ and not updating the discriminator weights, hence there is only EMD reconstruction loss for the generator.
- Ours (GT+EMD+GAN), in contrast, retains the adversarial training.

Table 5 shows the quantitative comparison results, we can see that, when the ground truth point sets are available, Ours (GT+EMD) produces better results than Ours (GT+EMD+GAN). Our explanation for why adopting adversarial training here leads to worse results is that: when the ground truth is available, which is complete and contains all information for supervising the network, adding adversarial training will make the network much harder to train, as the network always gets punished by failing to fool the discriminator when it is actually transforming current output closer to the ground truth.

Table 5: Removal of adversarial training when training with ground truth leads to significant improvement.

|  | Ours (GT+EMD) | | | Ours (GT+EMD+GAN) | | |
|---|---|---|---|---|---|---|
| model | acc. | comp. | F1 | acc. | comp. | F1 |
| car | 93.5 | 92.8 | **93.1** | 80.5 | 77.9 | 79.2 |
| chair | 82.3 | 83.3 | **82.8** | 51.5 | 58.1 | 54.6 |
| plane | 95.6 | 94.8 | **95.2** | 91.4 | 86.3 | 88.8 |
| table | 81.2 | 81.4 | **81.3** | 37.9 | 39.3 | 38.6 |

## G  MORE STATISTICS FRO THE BASELINE COMPARISON

For the baseline methods comparison on 3D-EPN dataset, we also report the Chamfer distance (CD), Earth Mover's Distance (EMD) and Hausdorff Distance (HD, maximum of the two directional distances) between the ground truth and the completion in Table 6:

Table 6

|  | AE | | | EPN | | | PCN | | | Ours | | | Ours+ | | |
|---|---|---|---|---|---|---|---|---|---|---|---|---|---|---|---|
| model | CD | EMD | HD | CD | EMD | HD | CD | EMD | HD | CD | EMD | HD | CD | EMD | HD |
| boat | 0.0012 | 0.0530 | 0.0864 | 0.0009 | 0.0500 | 0.0562 | **0.0006** | **0.0437** | **0.0635** | 0.0011 | 0.0532 | 0.0857 | 0.0008 | 0.0455 | 0.0799 |
| car | 0.0019 | 0.0668 | 0.1093 | 0.0024 | 0.0744 | 0.0989 | **0.0005** | 0.0418 | **0.0648** | 0.0010 | 0.0434 | 0.0763 | 0.0007 | **0.0393** | 0.0677 |
| chair | 0.0031 | 0.1003 | 0.1374 | 0.0016 | 0.0704 | 0.0877 | **0.0009** | **0.0586** | **0.0832** | 0.0020 | 0.0773 | 0.1010 | 0.0015 | 0.0619 | 0.0915 |
| dresser | 0.0037 | 0.0985 | 0.1295 | 0.0027 | 0.0783 | 0.0963 | **0.0008** | 0.0545 | 0.0771 | 0.0019 | 0.0588 | 0.0833 | 0.0011 | **0.0482** | **0.0734** |
| lamp | 0.0026 | 0.0857 | 0.1092 | 0.0038 | 0.0966 | 0.1154 | **0.0013** | **0.0692** | **0.0890** | 0.0023 | 0.0848 | 0.1073 | 0.0018 | 0.0729 | 0.1002 |
| plane | 0.0004 | 0.0346 | 0.0591 | 0.0060 | 0.0943 | 0.1255 | **0.0002** | **0.0308** | **0.0394** | 0.0004 | 0.0338 | 0.0545 | 0.0005 | 0.0405 | 0.0685 |
| sofa | 0.0030 | 0.0792 | 0.1232 | 0.0045 | 0.0880 | 0.1093 | **0.0008** | **0.0494** | **0.0651** | 0.0026 | 0.0655 | 0.0928 | 0.0012 | 0.0536 | 0.0958 |
| table | 0.0044 | 0.0886 | 0.1518 | 0.0014 | 0.0681 | 0.0975 | **0.0010** | **0.0603** | **0.0968** | 0.0026 | 0.068445 | 0.1071 | 0.0021 | 0.0681 | 0.1224 |

## H  USER STUDY ON REAL-WORLD DATA COMPLETION

We also conducted a user study on the completion results of the real-world scans, where given a partial input users are required to pick the most preferable completion among EPN, PCN and our results. In total, we received 1,000 valid user selections and report the preference (in percentage of the total selections) of each method in the user study. From Fig. 7 we can see that in over half (52%) of the selections, our completion results are selected as the best completion, while PCN completion results are better in 45% of the selections.

Interestingly, although our method outperforms other methods in the user study, our method clearly does not hold a dominant position in this user study. After a more in-depth analysis of the user selections, we found that users intend to pick the completion in which the partial input is *embedded*, which can be formulated as the Hausdorff distance from the partial input to the completion, and the supervised method PCN preserves the input point cloud in its completion output as this always minimizes the distance loss. The plausibility of the completion result is usually neglected by users,

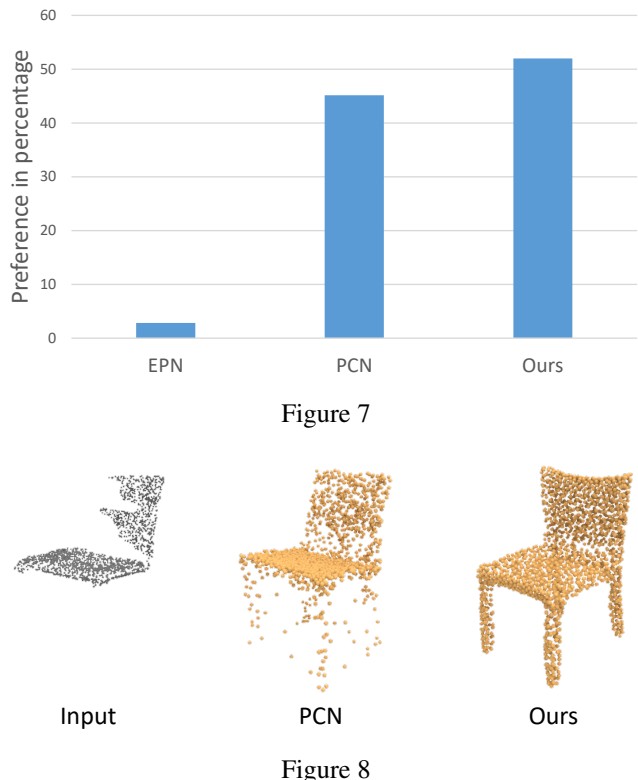

Figure 7

Figure 8

while the plausibility and the Hausdorff distance from the partial input to the completion are both considered as two trade-off terms in the objective function of our completion generator. Fig. 8 gives a typical example, where the PCN completion without clear chair structure is often picked as better completion while our completion tries to trade-off between the HL and the plausibility.

## I  GENERALIZATION TO UNSEEN CLASSES

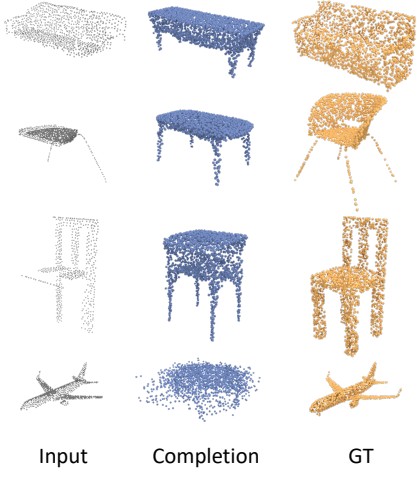

Figure 9

Our method does generalize to unseen objects from the same category in the test set, which is demonstrated in the experimental results section. However, our method intuitively should not generalize to classes that are not seen during the training, as the autoencoder, which is a fundamental component

in our network, does not generalize to unseen classes. We present the qualitative results of applying our table completion network on chair and airplane class. We can see that from Fig. 9, on the unseen chair class, which shares similar structure with table, our table completion network can produce some reasonable structures, but is unable to complete with a seat back as the autoencoder does not have such capability; on the unseen airplane class, which is rather dissimilar to table class, our table completion network failed to complete the partial airplanes.

