# OpenReview forum: "Unpaired Point Cloud Completion on Real Scans using Adversarial Training"
_ICLR.cc/2020/Conference — Accept (Poster)_

### Official Review · AnonReviewer1 · 2019-10-24
**Official Blind Review #1**

**Rating:** 6

**Review:**

This manuscript focuses on reconstructing 3D shapes from point clouds, with applications for instance to 3D scanners. The contribution builds on an adversarial formulation of the reconstruction, as in GANs. The method uses an encoder to map the observed noisy set of points into a lower-dimensional latent space and a decoder for the inverse mapping. The training loss is based on an Earth Mover's Distance between points. The training is done with an adversarial (min-max) strategy, that seeks to align the behavior of the encoder / decoder across a clean complete dataset and a partially-observed noisy one, with a Hausdorff distance loss, to cater for partial matching. The method is benchmarked on simulated and real data. It outperforms the state of the art for unsupervised settings (no known reconstructions) but for supervised settings it is slightly below the PCN approach.

I am not an expert of the application setting, hence I do not have many comments. The methods are well formulated and make sense. The experiments show the interest of the contributed method.

One thing that I do wonder is: given that the supervised approaches work better, but that there is not always data available for supervising, could a transfer learning approach be developed, adapting a supervised problem to a non supervised problem.

One question: how are the hyper-parameters set in the unsupervised setting? Appendix D details the model selection procedure: using a validation set and select the model that gives the best f1 score. I do not understand how this can be done without ground truth.

**Experience Assessment:**

I do not know much about this area.

**Review Assessment: Checking Correctness Of Derivations And Theory:**

I assessed the sensibility of the derivations and theory.

**Review Assessment: Checking Correctness Of Experiments:**

I assessed the sensibility of the experiments.

**Review Assessment: Thoroughness In Paper Reading:**

I read the paper at least twice and used my best judgement in assessing the paper.

---

> ### Author Response · Authors · 2019-11-12
> **Author's reply to Reviewer#1**
>
> We thank Reviewer#1 for valuable comments.
>
> Q: “how are the hyper-parameters set in the unsupervised setting? Appendix D details the model selection procedure: using a validation set and select the model that gives the best f1 score. I do not understand how this can be done without ground truth.”
>
> A: We thank the reviewer for this comment. The model selection procedure described in the appendix violates the unsupervised training setting, in which the ground truth is not available in both training and validation set. A common and straightforward strategy should be using the latest/last model saved from the training.
> We double-checked all the tables presented in the paper and updated Table 2, Table 4 and Table 5 (in the appendix), the section of model selection has also been removed from the appendix. We can see from the updated tables that there is only a minor difference between the two model selection procedures. In addition, all numbers of our method reported in other tables were produced by using the last-model selection strategy.

---

### Official Review · AnonReviewer3 · 2019-10-27
**Official Blind Review #3**

**Rating:** 6

**Review:**

The paper addresses the task of point cloud completion within an unpaired setting, where explicit correspondences between the partial and the complete shapes is not given. The setting represents significant interest in practice, e.g. in autonomous driving applications, where the precise completions of scanned objects, e.g. surrounding cars, are not necessary.

The authors propose to use three models, wherein two models are point autoencoders in the style of [2], obtaining the two spaces of latent codes for the partial and the complete shapes, respectively. The third model learns a mapping between the two latent spaces in an adversarial way. While the idea of doing the unpaired shape completion has been known since the introduction unpaired image-based methods (e.g., [1]), the application is novel and the methods are formulated using the language known in the point cloud learning literature (e.g., EMD losses and point-based autoencoders).

Regarding the evaluation procedure, the authors demonstrated convincingly that the proposed approach is feasible. However, I believe for shape completions with ground-truth labels more quality measures may be used, such as the Chamfer distance, Hausdorff distance, or Earth Mover’s distance (which is actually optimized by the authors), as shown in literature on point cloud upsampling [3], which is a related task.

Overall, I believe that the paper does a good job of combining the established components into something new and useful, particularly, the problem is well-defined, the method is intuitive and extends the state-of-the-art, and the evaluation looks convincing. The paper is well-written and easy to follow, too.


[1] Zhu, J. Y., Park, T., Isola, P., & Efros, A. A. (2017). Unpaired image-to-image translation using cycle-consistent adversarial networks. In Proceedings of the IEEE international conference on computer vision (pp. 2223-2232).
[2] Achlioptas, P., Diamanti, O., Mitliagkas, I., & Guibas, L. (2018, July). Learning Representations and Generative Models for 3D Point Clouds. In International Conference on Machine Learning (pp. 40-49).
[3] Lequan Yu, Xianzhi Li, Chi-Wing Fu, Daniel Cohen-Or, and Pheng-Ann Heng. Pu-net: Point cloud upsampling network. In Conference on Computer Vision and Pattern Recognition (CVPR), pp. 2790–2799, 2018b.

**Experience Assessment:**

I have published one or two papers in this area.

**Review Assessment: Checking Correctness Of Derivations And Theory:**

I assessed the sensibility of the derivations and theory.

**Review Assessment: Checking Correctness Of Experiments:**

I assessed the sensibility of the experiments.

**Review Assessment: Thoroughness In Paper Reading:**

I read the paper at least twice and used my best judgement in assessing the paper.

---

> ### Author Response · Authors · 2019-11-12
> **Author's reply to Reviewer#3**
>
> We thank Reviewer#3 for valuable comments.
>
> Q: "I believe for shape completions with ground-truth labels more quality measures may be used, such as the Chamfer distance, Hausdorff distance, or Earth Mover’s distance (which is actually optimized by the authors), as shown in literature on point cloud upsampling [3], which is a related task."
>
> A: We have added a new table for the baseline methods comparison using these metrics, please refer to Table 6 in the appendix of the paper.

---

### Official Review · AnonReviewer4 · 2019-11-03
**Official Blind Review #4**

**Rating:** 6

**Review:**

This paper proposes a new method for making 3D point clouds by automatically completing 3D scans. It does not require paired data samples for training which makes it possible to train it on real data instead of synthetic data. The authors use a generative adversarial network (GAN) to “generate” complete point clouds from noisy or partial point clouds obtained by 3D scanning. The generator learns to perform mapping from point set manifold of scanned noisy and partial input X_r to manifold of clean shapes X_c. The discriminator tries to tell between encoded clean shapes (synthetic data point clouds) and mappings of noisy input (point clouds from real-life data 3D scans).

An encoder-decoder network similar to those in Achlioptas et al. (2018) and Qi et al. (2017a) is trained to transform original point clouds to a low-dimensional latent space prior to training the GAN. The authors find that using the encoder-decoder trained on clean shape data even for noisy input yields better results.
One of the issues of completing noisy and partial scans is that the desired complete scan can have a very different shape compared to the noisy input. The generator can map latent vectors to any points on the target manifold which allows it to generate shapes that are far different from the original inputs. In order not to generate random clean shapes, the authors add a reconstruction loss to the generator which encourages it to preserve the partial shape of the input end reconstruct it in the completed clean shape. The choice of Hausdorff distance for reconstruction loss is sound and the ablation study confirms it.

The authors perform rather extensive experimental evaluation of their proposed method. They perform qualitative and quantitative analysis on several datasets, both real-life and synthetic ones. The proposed method outperforms existing methods in real-life data scenario. On the synthetic dataset (3D-EPN), the quantitative results are not as good as those of PCN (which is a supervised method, unlike the proposed method), but the qualitative results look plausible and comparable to PCN results. In terms of plausibility score, the proposed method outperforms existing methods in all experiments, which is probably thanks to its objective - map the input into the latent space of clean and complete shapes. However, plausible looking point clouds do not necessarily have to precisely match the input, which is the objective of 3D scan point cloud completion.
The ablation study also confirms the effectiveness of individual parts in the proposed method.
The main contribution of this paper is training with unpaired data, which enables training on real-life data, leading to better results on real-life scans. While I understand the difficulties, I believe it would be better to try to focus more on real-life data in the evaluation. There is only one experiment with quantitative analysis on real-life data in the paper. Supporting that with a visual Turing test would have been great.

Questions raised:

In 4.3, you say that “ground truth complete scans are not available for training.” How do you then train the supervised methods? Are they trained on 3D-EPN? In that case, is your proposed method also trained on 3D-EPN? If your method is the only one trained on your synthetic data (dataset D), which is also used for testing, then I do not think you could claim that your method is better at dealing with different data distributions. Please make clear in the comments what data you use for training in this experiment.

The meaning of section 4.4 is not very clear to me. If my understanding is correct, when completing noisy or partial point clouds, low diversity in results is better than high diversity because it is a task of repairing the input, which only has one correct result. Are you trying to say in this section that your method yields more consistent results with lower diversity than the method from previous work? If that is the case, you should consider rewriting that part to make it clearer to readers.

The PCN paper (Yuan et al., 2018) shows that PCN can generalize and complete point clouds of objects unseen during training very well. Unfortunately, this paper does not discuss performance on objects of unseen classes. Were such experiments considered? It would be beneficial to do a qualitative analysis of performance on unseen objects.

The description of the left table in Table 1 first says that it shows performance on real-life scans, but performance on synthetic data also appears there. Shouldn’t it rather say that it is plausibility comparison on real-life scans and synthetic data?

Summary:

The proposed method is easy to understand, exploits recent progress in generative models, and allows training on real-life scans as it does not require paired training data.
The paper does not contain much algorithmical novelty and mostly combines existing methods to solve the problem of obtaining clean and complete point clouds from real 3D scans. However, the ablation study shows that adding a good reconstruction loss to the generator is crucial for the performance.
The main strengths of this paper are extensive experimental evaluation using both quantitative and qualitative analysis, and significant improvement in performance on real-life data over previous works.
The main weakness is limited novelty in terms of the techniques used in the proposed method.

Additional comments that do not affect the review decision:

Citations in the text have to be revised and correctly put into parentheses where necessary. E.g., on page 2: “Since its introduction, GAN Goodfellow et al. (2014) has been used...” should be rewritten to “Since its introduction, GAN (Goodfellow et al., 2014) has been used...”

Table 2 is the only table where the best results are not highlighted by bold text. It is also the only table where the proposed method is not the best performing method as it loses to PCN. I apologize if it is just a pure coincidence but it seems as if the authors did not want to draw attention to the fact that their proposed method loses to an existing method in that experiment. I believe that you should be fair and highlight best results in all tables.


**Experience Assessment:**

I have read many papers in this area.

**Review Assessment: Checking Correctness Of Derivations And Theory:**

N/A

**Review Assessment: Checking Correctness Of Experiments:**

I carefully checked the experiments.

**Review Assessment: Thoroughness In Paper Reading:**

I read the paper thoroughly.

---

> ### Author Response · Authors · 2019-11-12
> **Authors' reply to Reviewer#4 -- main questions**
>
> We thank Reviewer#4 for valuable comments and address all questions in the following:
>
> Q: “...Supporting that with a visual Turing test would have been great. “
> A: As suggested, we conducted a user study on the completion results of real-world scans, where given a partial input users are required to pick the most preferable completion among EPN, PCN and our results. In total, we received 1,000 valid user selections and report the preference (in percentage of the total selections) of each method. We observe that in over half (52%) of the tests, our completion results are selected as the best completion, while PCN completion results are better in 45% of the selections.
> Interestingly, although our method outperforms other methods in the user study, our method clearly does not hold a dominant position in this user study. After a more in-depth analysis of the user selections, we found that users intend to pick the completion in which the partial input is embedded, which can be formulated as the Hausdorff distance from the partial input to the completion, and the supervised method PCN preserves the input point cloud in its completion as this intuitively minimizes the distance loss. The plausibility of the completion result is usually neglected by users while the plausibility and the Hausdorff distance are both considered as two trade-off terms in the objective of our generator. The Fig.8 in the appendix gives a typical example, where the PCN completion without clear chair structure is often picked as better completion while our completion tries to trade-off between the HL and plausibility.
> This user study has been added to the appendix.
>
> Q: “In 4.3, you say that “ground truth complete scans are not available for training.””
> A: In the setting of data discrepancy experiment, we assume no ground truth is available for training in our synthetic data (similar to real-world data, no ground truth annotation exists.), so only our method was trained with our synthetic data as only ours can be trained without ground truth. PCN and EPN were trained with 3D-EPN simulation data.
> The purpose of this experiment is to show how supervised methods suffer when the simulated training data is of different distribution to the test data. As the real-world data (which obviously has different data distribution to simulated scans) does not have ground truth for quantitative evaluation, we take our synthetic data as the test data of different data distribution, where we have ground truth only for quantitative evaluation to a) quantitatively measure the performance of supervised methods under data distribution discrepancy between its training and test data; b) quantitatively show our method performs well when ground truth is unavailable as ours does not require ground truth for training.
> Lastly, supervised methods assume its training and test data are drawn from the same data distribution, so the trained model can generalize to test data and outperforms unsupervised methods with proper training. This has already been demonstrated in Sec. 4.2, where we train all methods on 3D-EPN training data and test on 3D-EPN test data.
>
> Q: “The meaning of section 4.4 is not very clear to me...”
> A: In traditional content generation problem, higher diversity is always better than lower diversity. However, in our completion problem, the diversity of the completion results should match the diversity of the ground truth point clouds. So we present a) the JSD score between the ground truth point cloud set and our completion point cloud set, to show how close the diversity of our completion results is to that of the ground truth; b) And also the JSD score between ground truth point cloud set and extremely mode-collapsed point cloud set, which can be taken as a reference to show how far our completion is from the mode-collapse issue. The mode-collapse issue is alleviated by driving our network to match different input scans and thus our network does not output an *average* point cloud for completion.
> We have revised Sec. 4.4 to make it clearer.
>
> Q: “...this paper does not discuss performance on objects of unseen classes...”
> A: Our method does generalize to unseen objects from the same category in the test set, which is demonstrated in Sec. 4.2. However, our method intuitively should not generalize to classes that are unseen during the training, as the fundamental autoencoder does not generalize to unseen classes. We present the qualitative results of applying our table completion network on chair and airplane class. We can see in Fig.9 in the appendix, on the unseen chair class, which shares similar structure with table, our table completion network produces some reasonable structures, but is unable to complete with a seat back as the autoencoder has no such capability; on the unseen airplane class, which is rather dissimilar to table class, our table completion network failed to complete.
> This experiment has been included in the appendix.

---

> ### Author Response · Authors · 2019-11-12
> **Authors' reply to Reviewer#4 -- other corrections suggested**
>
> The following corrections as suggested have also been done in the paper:
>     1. The caption of Table 1 has been corrected.
>     2. Some citations have been corrected using the proper format.
>     3. The highest results are highlighted in Table 2 using bold text.

---

### Decision · Program_Chairs · 2019-12-19

**Decision:**

Accept (Poster)

**Comment:**

This paper presents an unsupervised method for completing point clouds obtained from real 3D scans based on GAN. Generally, the paper is well-organized, and its contributions and experimental supports are clearly presented, from which all reviewers got positive impressions.
Although the technical contribution of the method seems marginal as it is essentially a combination of established methods, it well fits in a novel and practical application scenario, and its useful is convincingly demonstrated in intensive experiments. We conclude that the paper provides favorable insights covering the weakness in technical novelty, so I’d like to recommend acceptance.